# A Randomized Clinical Trial to Assess the Efficacy of a Psychological Treatment to Quit Smoking Assisted with an App: Study Protocol

**DOI:** 10.3390/ijerph19159770

**Published:** 2022-08-08

**Authors:** Ana López-Durán, Elisardo Becoña, Carmen Senra, Daniel Suárez-Castro, María Barroso-Hurtado, Carmela Martínez-Vispo

**Affiliations:** 1Smoking and Addictive Disorders Unit, University of Santiago de Compostela, 15782 Santiago de Compostela, Spain; 2Department of Clinical Psychology and Psychobiology, University of Santiago de Compostela, 15782 Santiago de Compostela, Spain

**Keywords:** smoking cessation, smartphone app, randomized controlled trial, relapse prevention

## Abstract

Numerous studies have shown the efficacy of smoking cessation interventions. However, some challenges, such as relapse rates, remain. The availability of information technologies (ICTs) offers promising opportunities to address such challenges. The aim of this paper is to describe the protocol followed to assess the efficacy of a face-to-face cognitive–behavioral intervention for smoking cessation using a smartphone application as a complement, compared with a control group. A single blind, two-arm, randomized controlled trial is proposed (NCT04765813). The participants will be smokers over 18 years old, who smoke at least eight cigarettes per day. Participants will be randomized to one of two conditions, using a 1:1 allocation ratio: (1) cognitive–behavioral smoking cessation treatment along with an App with active therapeutic components (*SinHumo* App); or (2) cognitive–behavioral treatment along with the use of a control App (without active components). The experimental App will be used during the eight treatment sessions and for 12 months after the end of treatment. The primary outcome measures will be 7-days point-prevalence abstinence at 12-months follow-up. We expect the experimental App to obtain higher abstinence rates at the end of treatment and at one-year post-treatment follow-ups and lower relapse rates, compared to the control App.

## 1. Introduction

Smoking remains the leading avoidable cause of disability, mortality, and morbidity worldwide [1], and it is responsible for more than 8 million deaths around the world every year [2]. Despite the consistent evidence demonstrating that tobacco use is a significant health-related risk factor, 28% of adults smoked in the European Region in 2020 [2], constituting a significant public health concern. The most common health complications related to smoking are cancer, respiratory disease, and cardiovascular disease [3]. Smoking is also associated with certain mental disorders, such as anxiety, depression, and schizophrenia [4]. In addition, smoking related diseases lead to significant health care costs, accounting for 5.7% of global health care expenditure, which shows the huge economic impact of smoking worldwide [5].

Psychological treatments (i.e., cognitive–behavioral approaches, behavioral counseling) for smoking cessation are first-line treatments that have proven efficacy to quit smoking among different populations [6]. A systematic review carried out by Stead et al. [7] concludes that group behavioral interventions for smoking cessation, when compared to self-help programs, brief advice, or no intervention, obtain better results, with a risk ratio (RR) of 1.88, 1.25, and 2.6, respectively. However, some challenges remain, such as treatment accessibility [8] or relapse rates after quitting [9,10]. In fact, a significant percentage of smokers who achieve abstinence relapse within weeks or months of completing treatment [11]. Some studies show that twelve months after the end of a smoking cessation treatment, relapse rates are found to be 70% in psychological interventions [12], and around 80% in pharmacological interventions [10].

Nowadays, the availability of information technologies (ICTs) (i.e., the Web, smartphones) offers promising opportunities to address such challenges. The use of ICTs in the field of health has grown and developed significantly, demonstrating their utility in several areas, such as diabetes management, health-promoting behaviors (i.e., physical activity), or mental health [13,14]. For instance, in a systematic review, Erbe et al. [15] concluded that Internet-based interventions to support face-to-face treatment for adults with mental disorders decrease the likelihood of treatment dropout, help maintain therapeutic changes, and increase abstinence rates in people with substance use disorders.

In this context, the use of mHealth (mobile health) is becoming increasingly promising. mHealth is defined as the use of mobile devices, such as mobile phones or personal digital assistants (PDA), as a means to support or deliver health-related services [16]. Specifically, in recent years, a major development of smartphone Apps for health promotion and maintenance has been observed along with the rapid increase of mobile phone ownership and usage worldwide [17]. These advances can also be seen in the field of psychological intervention with the development of smartphone apps that target different problems and mental disorders. For instance, several studies have suggested that using a mobile App to complement face-to-face interventions addressing addictive disorders could improve abstinence outcomes [18]. In fact, a study conducted by Gustafson et al. [19], comparing the efficacy of two cognitive–behavioral treatments for alcohol consumption (with and without the use of an App), found positive outcomes during the eight months of the intervention and the four-months follow-up in the App condition. It was also considered that the use of Apps could be useful as an adjunct to psychological smoking cessation treatment [20]; in particular, to achieve abstinence and prevent relapse [21,22].

mHealth Apps for smoking cessation are being widely developed, and although they offer promising results, they are still modest [23]. These Apps can be classified as standalone general apps for smoking cessation without any face-to-face contact with health-care professionals, or smoking cessation apps that are designed to be used as a complement to face-to-face intervention [21]. Recent reviews point out that smoking cessation apps combined with face-to-face intervention improve abstinence outcomes [24]. However, the most increased development has been for standalone self-help Apps, and scarce studies have been conducted to assess the efficacy of a face-to-face treatment combined with an App compared to face-to-face treatment alone. Additionally, recent systematic reviews have highlighted the need for well-powered randomized controlled trials to examine the efficacy of smoking cessation apps [21,25]. Therefore, examining whether using a mobile App with a face-to-face psychological smoking cessation treatment could improve abstinence outcomes is warranted.

The main aim of the present randomized controlled trial is to examine whether the SinHumo smartphone application (iOS and Android), used as a complement of a face-to-face cognitive–behavioral smoking cessation treatment, improves smoking cessation rates at the end of treatment, and at 3-, 6-, and 12-months follow-up, compared with a control group which receives the same smoking cessation treatment and a control App (without active components). The main hypothesis is that combining a cognitive–behavioral psychological treatment to quit smoking and an App with active therapeutic components will obtain higher abstinence rates at the end of treatment and the 12-month follow-up period and lower relapse rates compared to the control group.

## 2. Materials and Methods

### 2.1. Research Design

The proposed study is a two-arm, single-blind, randomized controlled trial (NCT04765813) to assess the efficacy of a cognitive–behavioral intervention for smoking cessation complemented by an App. The overall study design is summarized in Figure 1.

### 2.2. Recruitment

Participants will be recruited through the media, publications on the Smoking Cessation Unit’s social networks (Facebook and Instagram), posters in health-care centers, word of mouth, or referred to the Unit by their primary care physician or other specialized services of the health-care system. Independently of their origin (patients referred by their primary care physician, word of mouth, or social media), all potential participants have to personally contact the trial staff by telephone or email. Before participants enroll in the study, written informed consent will be obtained. Then, the different phases of the study (assessment, treatment, and follow-ups) will be carried out.

### 2.3. Participants

Sample size calculation was performed using the software developed by Wang and Ji [26]. To detect a 20% difference (50% abstinence in the experimental group vs. 30% in the control group) between conditions at the 12 months follow-up (90% power at an alpha of 0.05), a minimum of 99 participants per group will be required. With an estimated attrition of 25%, 264 participants (132 per group) are required.

To be included in the study, participants should meet the following inclusion criteria: (1) 18 years of age or older; (2) smoke at least six cigarettes per day; (3) desire to participate voluntarily in the smoking cessation treatment; (4) correctly fill out all the pre-treatment assessment questionnaires; (5) able to provide written informed consent; (6) currently own a valid email address; and (7) having a smartphone with an Internet connection (Android or iOS) and being willing to use it during treatment.

Exclusion criteria include: (1) having a diagnosis of a severe mental disorder (bipolar disorder and/or psychotic disorder); (2) having a substance use disorder (alcohol, cannabis, cocaine, heroin), different from a tobacco use disorder; (3) smoke rolling tobacco, snuff, cigars, little cigars, or other tobacco products (i.e., e-cigarettes); (4) have participated in an effective psychological treatment to quit smoking during the previous 12 months; (5) have received an effective pharmacological treatment to quit smoking in the previous 12 months (nicotine gum or patches, bupropion, varenicline); (6) have a physical pathology involving life-threatening risks for the person which would require immediate intervention in an individual format (e.g., recent myocardial infarction, pneumothorax); and (7) have a visual impairment that impedes the use of the App.

### 2.4. Treatment Interventions

#### 2.4.1. Cognitive Behavioural Smoking Cessation Intervention and SinHumo App (Experimental Condition)

This intervention is a multicomponent cognitive–behavioral treatment (“Programa para Dejar de Fumar”) [27], which has been manualized and used in different contexts (i.e., hospitals, private companies) in Spain. The description of the main intervention has been previously published [28,29]. The main therapeutic components are: (1) treatment contract; (2) self-report and graphic representation of cigarette consumption; (3) information about tobacco; (4) nicotine fading (gradual reduction of nicotine and tar ingestion); (5) stimulus control; (6) activities to prevent withdrawal syndrome; (7) relapse prevention strategies (including assertion training, problem-solving training, changing tobacco-related misconceptions, management of anxiety and anger, exercise, weight control, and self-reinforcement); and (8) components of Behavioural Activation (BA), including an analysis of the relationship between behavior and mood, identifying situations and behaviors that decrease mood, identifying avoidance behaviors, identifying rumination and worry, self-reporting pleasant daily activities, pleasant activity scheduling to increase engagement in rewarding activities, and reducing patterns of behavioral avoidance (Table 1). 

This intervention is applied in a group format and consists of 8 one-hour sessions once a week. Due to the COVID-19 pandemic, these sessions will be conducted through video calls (i.e., Microsoft Teams).

Along with the intervention, participants will be provided with an App with active therapeutic components, which will be used during the treatment sessions (Phase 1) and the follow-up period (Phase 2):

-During Phase 1 (Table 1), the App will have the following content: smoking self-monitoring tool, cigarettes consumption graph, a list of personal reasons to quit smoking, setting weekly cigarette reduction goals tool (to nicotine fading), stimulus control tool (set weekly non-smoking situations), deep breathing videos, self-monitoring of BA activities tool, inter-session motivational and weekly reinforcement notifications of goal achievements, and access to the written materials in pdf format provided during the face-to-face sessions (Figure 2).-During Phase 2 (Table 2), the App content will differ depending on the participants’ smoking status, which are defined as follows: (a) abstinents, as those participants who report being abstainers; (b) relapsed, as those participants who report abstinence at the end of treatment, but who relapsed during the follow-up period; and (c) smokers, as those participants who never quit smoking during treatment and follow-ups. The specific components are detailed below:
(a)*Abstinent*: gains and achievements made since quitting smoking (number of days without smoking, saved money, time gained, and physical improvements), behavioral and cognitive recommendations for abstinence maintenance, strategies for coping with cravings (distraction tools, such as gifs, games, or videos; and social-support-seeking strategies, such as access to an emergency contact list), motivational strategies (list reasons to remain abstinent, personal video recording focused on reasons for quitting, and videos of close friends/relatives motivating them to remain abstinent), access to the materials in pdf format that were provided during the sessions, and the possibility of self-reporting tobacco use if a lapse or relapse occurs (Figure 3). Likewise, they will receive notifications during this follow-up phase according to gains and achievements as they remain abstinent through the one-year follow-up.(b)*Relapsed*: behavioral and cognitive advice to quit (tips), the possibility of setting a new quit date in a calendar, strategies to cope with cravings (distraction tools, such as gifs, games, or videos; and social-support-seeking strategies, such as access to an emergency contact list), motivational strategies (list of personal reasons for quitting, personal video with reasons for quitting, and videos of close friends/relatives motivating them to quit), access to the materials in pdf format that were provided during the sessions, and the possibility of self-reporting tobacco use. They will receive notifications to encourage and support a new quit attempt during this follow-up phase (Figure 4).(c)*Smokers*: behavioral and cognitive advice to quit (tips), the possibility of setting a quit date in a calendar, notifications consisting of strategies to be prepared for cessation, strategies to cope with cravings (distraction tools, such as gifs, games, or videos; and social-support-seeking strategies, such as access to an emergency contact list), motivational strategies (list of personal reasons for quitting, personal video with reasons for quitting, and videos of close friends/relatives motivating them to quit), access to the materials in pdf format that were provided during the face-to-face sessions, and the possibility of self-reporting tobacco use. Likewise, they will receive notifications to encourage a quit attempt during this follow-up phase (Figure 5).

The SinHumo App was developed by a team of psychologists, researchers, and computer programmers as an evidence-based App to complement a cognitive–behavioral intervention. Regarding the development of the SinHumo App, firstly, a review of available smoking cessation Apps literature was conducted to examine the main features and outcomes of existing Apps to quit [21]. Then, expert meetings were scheduled to design the components and features of the App, based on the extracted information. The App components integrate practice guidelines for treating tobacco use and dependence [29], and principles of behavior change techniques [30]. Finally, the usability and performance of the application was tested in a group of smokers.

#### 2.4.2. Cognitive Behavioural Smoking Cessation Intervention and Control App (Control Condition)

The cognitive–behavioral psychological treatment applied in this condition is identical to the one described above in the experimental condition, but with a control App (Table 1 and Table 2). This App will be used during the treatment sessions (Phase 1) and follow-up period (Phase 2), with only two basic contents: a smoking self-report tool, and access to the written materials in pdf format provided during the face-to-face sessions.

### 2.5. Procedures

One baseline assessment session will be conducted in an individual face-to-face interview using a video call. The researchers conducting the baseline assessments will be blind to group allocation, which will occur subsequently. The randomization will be performed according to a computer-generated allocation sequence (ratio: 1.1).

The cognitive–behavioral treatment will be delivered through video calls using Microsoft Teams. The smoking cessation treatment has been manualized to train the therapists and improve the intervention implementation. It includes a detailed session-by-session protocol and follow-up procedures. Study supervisors will assess fidelity by randomly visualizing treatment sessions in both arms.

Both conditions will be administered in eight weekly 60-min sessions. At the end of treatment (session 8), there will be a post-treatment assessment and face-to-face follow-ups through video calls at 3, 6, and 12 months. Trained therapists (Master level in clinical or counseling psychology) will conduct the assessment, intervention sessions, and follow-ups. 

This study was reviewed and approved by the Institutional Ethics Review Board of the University of Santiago de Compostela (USC-15/2020). The study is carried out in accordance with the Declaration of Helsinki. The potential participants will be asked to read and sign an informed consent form before any study procedures. The interested individuals will be initially screened for eligibility, and those who meet the inclusion criteria will be assessed. 

### 2.6. Measures

An online face-to-face structured interview will assess sociodemographic characteristics, tobacco-related information, smoking cessation treatment history, and depression treatment history. In addition, the following instruments will be used (Table 3 lists the measures collected at each time point):-*Smoking Habit Questionnaire* [30]. This questionnaire consists of 56 items gathering information about sociodemographic variables (gender, age, marital status, educational level) and tobacco use (i.e., number of cigarettes smoked per day).-*Fagerström Test for Cigarette Dependence* (FTCD) [31,32]. This scale assesses cigarette dependence with six items. Cronbach’s alpha coefficient was 0.60 in studies conducted in Spain [33].-*Nicotine Dependence Syndrome Scale* (NDSS) [34,35]. This questionnaire assesses nicotine dependence using a multidimensional conceptualization of substance dependence. The reliability of the general factor that evaluates nicotine dependence is adequate (Cronbach’s alpha 0.80).-*Structured interview for the assessment of the DSM-5 diagnostic criteria for tobacco use disorder* [36].-*Minnesota Nicotine Withdrawal Scale* (MNWS) [37]. This 8-item scale measures nicotine withdrawal symptoms (depression, insomnia, irritability/frustration/anger, anxiety, difficulty concentrating, restlessness, increased appetite/weight gain) and craving (smoking urgency). The reliability of this instrument is good (Cronbach’s alpha 0.85).-*Self-Efficacy/Smoking Temptation Scale* [38]. This 9-items instrument assesses the temptation to smoke in different situations: positive affect/social situations, positive affect situations, and habit situations.-*Questionnaire of Urgency to Smoke* [39]. This questionnaire consists of 10 items evaluating tobacco craving. This questionnaire has two factors: (1) the intention or desire to smoke, and (2) the expectations of negative reinforcement or improvements through smoking.-*Beck Depression Inventory II* (BDI-II) [40,41]. This self-report consists of 21-items assessing depressive symptoms during the last two weeks. The reliability of this instrument is good (Cronbach’s alpha 0.90).-*EuroQoL-5D Quality of Life Questionnaire* [42]. This is a standardized and generic instrument to assess health-related quality of life. It consists of (1) assessment of five dimensions: mobility, self-care, activities of daily living, pain/discomfort, and anxiety/depression; and (2) a visual-analog scale where the individual indicates the score (ranging from 0 to 100) that best represents his/her overall health status on the day of the interview.-*Questionnaire about the use of smartphones and Apps*. This is an ad hoc questionnaire collecting information about the use of smartphones (i.e., frequency, time of use, type of use) and Apps (i.e., frequency of use, hours of use, types of Apps).-*End-of-treatment questionnaire*. This self-report collects information about quit date, confidence in remaining abstinent, perceived social support, and physical and psychological improvements since the beginning of treatment.-*Client Satisfaction Questionnaire* [43]. This 8-item instrument measures overall satisfaction with treatment services.-*Satisfaction with App questionnaire*, in which data about satisfaction and usability are collected at the end of treatment and at 3-, 6-, and 12-months follow-up.-*Follow-up questionnaire*, in which abstinence and lapse/relapse data are collected at 3-, 6-, and 12-months follow-up.

### 2.7. Outcomes

The main smoking cessation outcomes were defined based on the updated recommendations by the Society for Research on Nicotine and Tobacco (SRNT) Treatment Research Network [44].

#### 2.7.1. Primary Outcome

The primary outcome was the self-reported seven-day point prevalence of complete smoking abstinence (“not even a puff”) at 12-months follow-up. 

#### 2.7.2. Secondary Outcomes

The secondary outcomes were:

The self-reported seven-days point prevalence of complete smoking abstinence (“not even a puff”) at the end of treatment (week 8), 3-, and 6-months follow-up;

The self-reported 30-days point prevalence smoking abstinence at 3-, 6-, and 12-months follow-up;

The self-reported prolonged smoking abstinence with lapses (no more than 5 cigarettes) from the end of treatment and each follow-up point assessment at 3-, 6-, and 12-months;

The reduction by 50% or more in the number of cigarettes smoked per day from baseline to the one-year follow-up.

We will also explore the adherence, satisfaction, usability of the App, and specific tool usage in relation to smoking cessation outcomes.

### 2.8. Data Management and Confidentiality

Data will be collected from the participants in an electronic format using a unique identification code number assigned to each participant for trial documents and electronic databases. All data will be kept in password-protected computer folders at the Smoking Cessation and Addictive Disorders Unit (University of Santiago de Compostela). Only authorized trial staff will have access to trial documents.

### 2.9. Ethical Principles

This study has been designed in accordance with the Declaration of Helsinki. The Bioethics Committee of the University of Santiago de Compostela approved this study (USC-15/2020), which is registered with the international standard randomized controlled trial number, NCT04765813 (www.clinicaltrials.gov accessed on 21 February 2021). The participants will be provided with both oral and written information regarding the study before obtaining their informed consent.

### 2.10. Trial Status

The present study started participants’ enrollment in September 2021, and the recruitment is ongoing.

### 2.11. Statistical Analysis

A descriptive analysis to summarize the characteristics of the total sample will be conducted. Comparisons between the two groups will be performed using an independent sample *t*-test for continuous variables (i.e., number of cigarettes smoked per day) and a chi-squared test for categorical variables (i.e., smoking status). Logistic regression analyses adjusted for baseline characteristics (i.e., sex, age, education level) will be carried out to examine whether abstinence rates differ significantly between conditions at the end of treatment, and at 3-, 6-, and 12-months follow-up. Mediation analysis will be conducted to analyze the role of different variables (i.e., depressive symptoms) in the relationship between treatment condition and smoking status. Multilevel and survival analysis will also be carried out to examine the relapse process. All data analysis will follow the intention-to-treat principle, which includes in the primary analysis all randomized participants [45]. A *p* value < 0.05 will be considered statistically significant.

The study results will be reported following the Consolidated Standards of Reporting Randomized Trials of Social and Psychological Interventions (CONSORT-SPI 2018) statements [46,47].

## 3. Discussion

The present study will assess whether adding an App to a cognitive–behavioural smoking cessation treatment will improve short- and long-term abstinence rates, and reduce relapse rates in a sample of treatment-seeking smokers using a randomized clinical trial. The experimental condition will be provided with an App with therapeutic components (i.e., motivational strategies, craving management tools) along with the intervention and during a one-year follow-up period. The control group will receive the same smoking cessation treatment along with a control App in which participants are provided with a set of pdf documents that expand on the topics addressed in each session and a smoking self-monitoring tool. Adding an App with active cognitive–behavioral components to support the quitting process to a face-to-face intervention and throughout a year follow-up is innovative and might result in greater abstinence rates and lower relapse rates. We hypothesize that using this App would increase treatment intensity and accessibility to psychological strategies to quit, supporting the participants’ smoking cessation process.

Nowadays, the frequent use of smartphones and Apps is part of most people’s daily lives, and this is a resource that is beginning to be used for interventions aimed to promote the health of patients. An App with active components used as a complement to a psychological treatment, such as the one mentioned above, would also gather information about task completion and the tools used [18], providing relevant information about the adherence to the intervention.

Interventions for smoking cessation should pay special attention to relapse prevention, as relapse is a barrier to any treatment that targets addictive behaviors, and one of the main reasons for reduced overall success rates [48,49]. One of the strengths of the present project is that the experimental App specifically targets relapse prevention in abstinent patients, and relapse management in Phase 2 during the 12 months after the last treatment session, a period of time when relapse is most likely to occur. Thus, the goal of this Phase 2 of the experimental App will be relapse prevention in a critical time period where most relapses occur [12].

Some potential limitations should be acknowledged. As noted earlier, due to the COVID-19 pandemic and resource limitations, abstinence will only be self-reported, and biochemical verification will not be performed. The biochemical verification of abstinence is desirable, as it increases scientific rigor, limiting the impact of misreporting smoking status [50]. However, the current circumstances limit the benefits of biochemical verification due to participant and staff safety, economic costs, and sample collection feasibility. Moreover, the use of video calls instead of in-person sessions might impact trial outcomes. Video calls attended at home might be perceived by some patients as being intrusive; moreover, minorities, lower-income, and older populations might have more difficulties to access virtual treatment [51]. Similar limitations may be encountered in regard to the characteristics of the users of mHealth Apps [52]. However, the strengths, such as cost-effectiveness, ease to deliver at a large scale, accessibility, and attractiveness [24], clearly overpower these limitations.

## 4. Conclusions

We expect that adding a smartphone App to a well-established behavioral smoking cessation treatment will improve smoking cessation outcomes at the end of treatment and will reduce relapse rates at follow-ups. These findings will have important clinical implications, potentially transforming the way that people interact with health-care services. Any public health effort that targets smoking cessation is essential to improve people’s health and quality of life, and to reduce public health-care expenditure.

## Figures and Tables

**Figure 1 ijerph-19-09770-f001:**
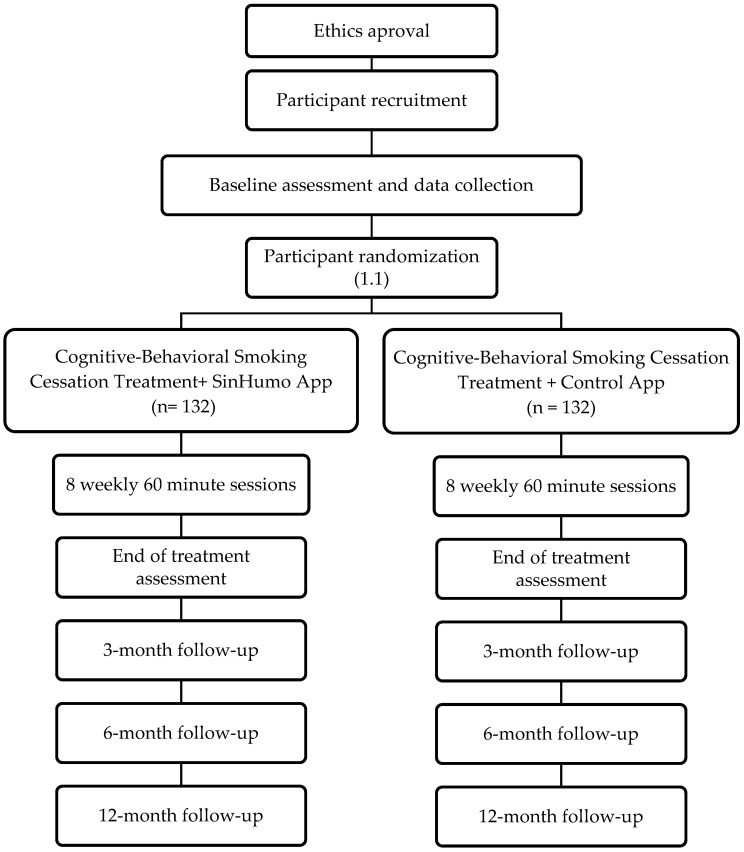
Consort Diagram of Trial Progression.

**Figure 2 ijerph-19-09770-f002:**
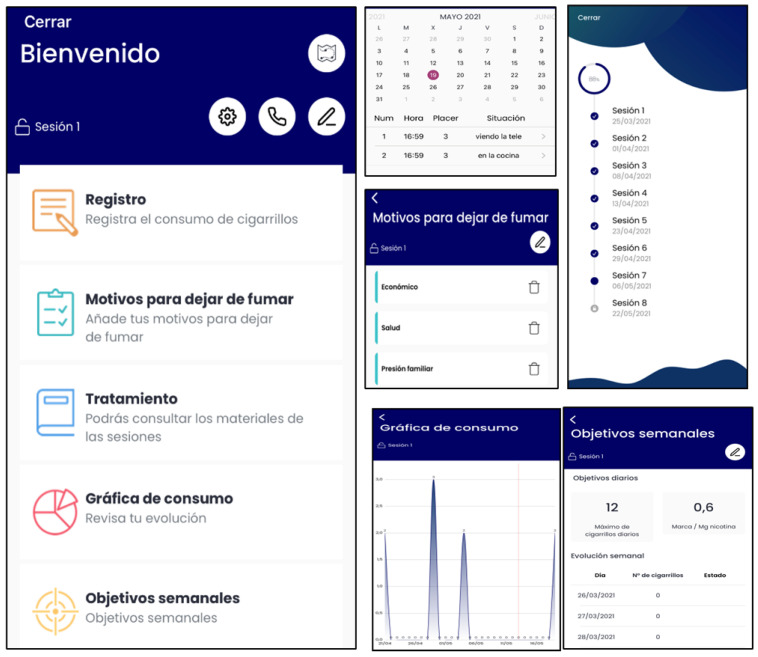
SinHumo App features during phase 1.

**Figure 3 ijerph-19-09770-f003:**
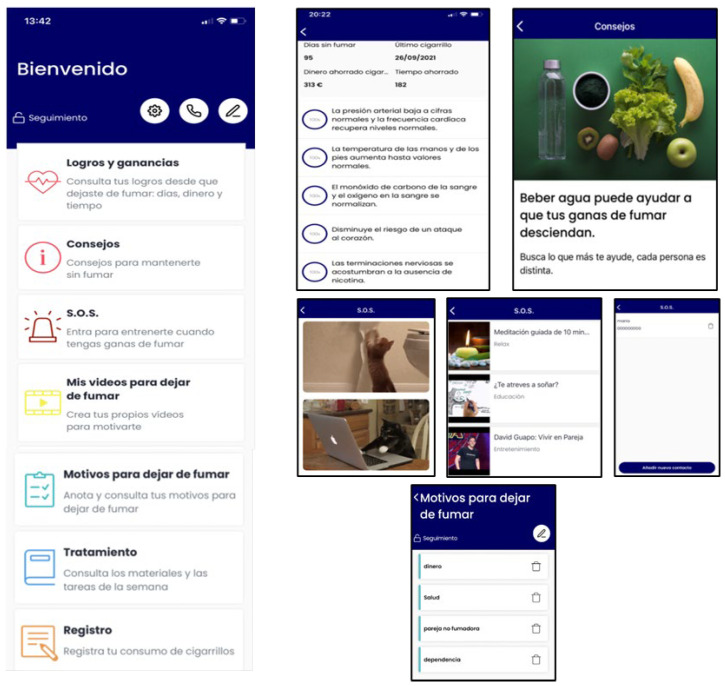
SinHumo App features during phase 2 for abstainers.

**Figure 4 ijerph-19-09770-f004:**
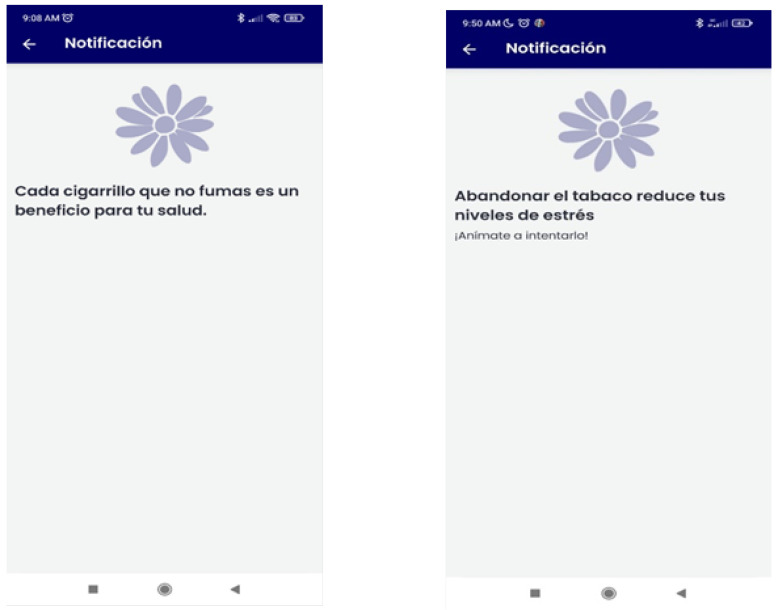
Motivational notifications samples during phase 2 for relapsers (on the **left**) and smokers (on the **right**) of the SinHumo App.

**Figure 5 ijerph-19-09770-f005:**
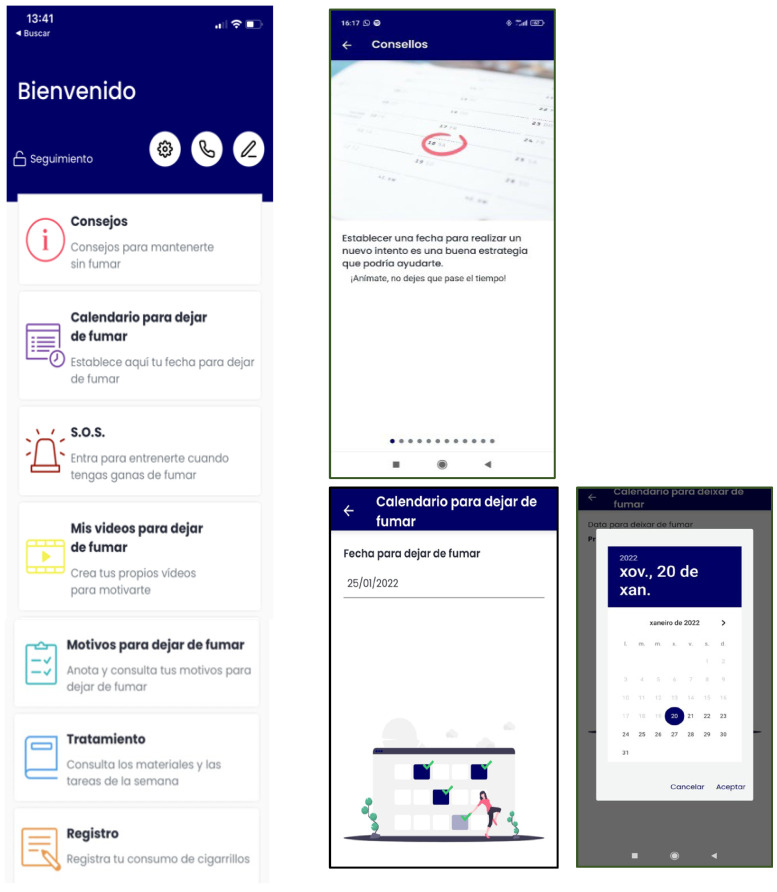
SinHumo App features during phase 2 for relapsers and smokers.

**Table 1 ijerph-19-09770-t001:** Summary of session-by-session outline of intervention procedures and App components (Phase 1) in the experimental and control conditions.

	Experimental Condition (SinHumo App)	Control Condition (Control App)
Session1	Overview of treatment and App usage Smoking cessation treatment rationale Review cigarette self-monitoring using the App (tracking cigarettes smoked, pleasure experienced, and smoking antecedents and consequences) Automatically generated cigarettes consumption graph available in the App Discussion of reasons for smoking and for quitting and completion tool available in the App named “reasons to quit” Discussion about smoking history and past quit experiences Explanation of written materials included in the App about tobacco, nicotine dependence, smoking health consequences, and cessation benefits Explain nicotine fading through brand change and set weekly cigarette reduction goals using the App tool Intersession activities: - Brand change - Communicate to at least one person (family, friend, coworker, etc.) that he/she is trying to quit smoking in the next 30 days - Not smoking more cigarettes than the average of those smoked the previous week - Leave a third of the cigarette without smoking - Refuse cigarette offers Intersession motivational notification through the App	Overview of treatment Smoking cessation treatment rationale Review cigarette self-monitoring using the App (tracking cigarettes smoked, pleasure experienced, and smoking antecedents and consequences) Discussion of reasons for smoking and for quitting Discussion about smoking history and past quit experiences Explanation of written materials included in the App about tobacco, nicotine dependence, smoking health consequences, and cessation benefits Explain nicotine fading through brand change Intersession activities: - Brand change - Communicate to at least one person (family, friend, coworker, etc.) that he/she is trying to quit smoking in the next 30 days - Not smoking more cigarettes than the average of those smoked the previous week - Leave a third of the cigarette without smoking - Refuse cigarette offers
Session 2	Check homework and nicotine fading compliance Continue smoking self-monitoring through the App and analyze smoking behavior during the week Automatically generated cigarettes consumption graph available in the App Discuss brand change difficulties New brand change and reduction of number of cigarettes to smoke the next week, setting the cigarette reduction goal using the App tool Review importance of social support Introduce stimulus control technique to remove situations conditioned to smoking using the App Nicotine withdrawal and strategies to avoid it Breathing exercises and relaxation techniques (practice as homework using the videos available in the App) Intersession motivational notification through the App Reinforcement notification of goal achievement through the App	Check homework and nicotine fading compliance Continue smoking self-monitoring through the App and analyze smoking behavior during the week Discuss brand change difficulties New brand change and reduction of number of cigarettes to smoke the next week Review importance of social support Introduce stimulus control technique to remove situations conditioned to smoking Nicotine withdrawal and strategies to avoid it Breathing exercises and relaxation techniques (practice such as homework)
Intervention Session 3	Check nicotine fading, cigarette reduction, and stimulus control compliance Check breathing exercises compliance and strategies to avoid withdrawal symptoms Continue cigarette self-monitoring through the App and analyze smoking behavior Automatically generated cigarettes consumption graph available in the App New brand change and reduce number of cigarettes to smoke the next week setting the cigarette reduction goal using the App tool Continue stimulus control technique using the App Explain written materials for weight control and exercise available in the App Continue with breathing exercises compliance and strategies to avoid withdrawal symptoms Rationale of mood influence in smoking cessation (written materials available in the App) Homework: daily activities self-monitoring using App tool Intersession motivational notification through the App Reinforcement notification of goal achievement through the App	Check nicotine fading, cigarette reduction, and stimulus control compliance Check breathing exercises compliance and strategies to avoid withdrawal symptoms Continue cigarette self-monitoring through the App and analyze smoking behavior New brand change and reduce number of cigarettes to smoke the next week Continue stimulus control technique Explain written materials for weight control and exercise available in the App Continue with breathing exercises compliance and strategies to avoid withdrawal symptoms Rationale of mood influence in smoking cessation (written materials available in the App) Homework: daily activities self-monitoring
Session 4	Check nicotine fading, cigarette reduction, and stimulus control compliance Check breathing exercises compliance and strategies to avoid withdrawal symptoms Continue smoking self-monitoring through the App and analyze smoking behavior Automatically generated cigarettes consumption graph available in the App New brand change and reduction of number of cigarettes to smoke the next week, setting the cigarette reduction goal using the App tool Continue stimulus control technique using the App Stress and anxiety management strategies Check activity scheduling through the App and encourage recognizing patterns of depressed behavior and the way in which engaging in enjoyable and important activities may impact their overall mood Homework: continue activity scheduling through the App, create a pleasant activities list and choose one to do during the week Intersession motivational notification through the App Reinforcement notification of goal achievement through the App	Check nicotine fading, cigarette reduction, and stimulus control compliance Check breathing exercises compliance and strategies to avoid withdrawal symptoms Continue smoking self-monitoring through the App and analyze smoking behavior New brand change and reduction of number of cigarettes to smoke the next week Continue stimulus control technique Stress and anxiety management strategies Check activity scheduling, and encourage recognizing patterns of depressed behavior and the way in which engaging in enjoyable and important activities may impact their overall mood Homework: continue activity scheduling, create a pleasant activities list, and choose one to do during the week
Session 5	Check nicotine fading, cigarette reduction, and stimulus control compliance using the App Check activity scheduling, pleasant activities list elaboration, and pleasant activity compliance using the App tools Continue smoking self-monitoring through the App and analyze smoking behavior Automatically generated cigarettes consumption graph available in the App New brand change and reduction of number of cigarettes to smoke the next week, setting the cigarette reduction goal using the App tool Management of anxiety and anger Self-reinforcing Changing tobacco-related misconceptions Problem-solving training Recognize avoidance behavior and impact on mood Activity scheduling through the App and engagement in two pleasant activities during the week Intersession motivational notification through the App Reinforcement notification of goal achievement through the App	Check nicotine fading, cigarette reduction, and stimulus control compliance Check activity scheduling, pleasant activities list elaboration, and pleasant activity compliance Continue smoking self-monitoring through the App and analyze smoking behavior New brand change and reduction of number of cigarettes to smoke the next week Management of anxiety and anger Self-reinforcing Changing tobacco-related misconceptions Problem-solving training Recognize avoidance behavior and impact on mood Activity scheduling and engagement in two pleasant activities during the week
Session 6	Quitting experience and withdrawal symptoms Discuss and plan for high-risk lapse and relapse situations Motivating factors for maintaining abstinence Benefits of quitting smoking Common barriers for maintaining abstinence Ruminative thoughts, smoking cessation process, and relapse Check activity scheduling and pleasant activity compliance using the App Activity scheduling through the App for the next week and engagement in 2 pleasant activities/week Intersession motivational notification through the App Reinforcement notification of goal achievement through the App	Quitting experience and withdrawal symptoms Discuss and plan for high-risk lapse and relapse situations Motivating factors for maintaining abstinence Benefits of quitting smoking Common barriers for maintaining abstinence Ruminative thoughts, smoking cessation process, and relapse Check activity scheduling and pleasant activity compliance Activity scheduling for the next week and engagement in 2 pleasant activities/week
Session 7	Quitting experience and withdrawal symptoms Discuss and plan for high-risk lapse and relapse situations Motivating factors for maintaining abstinence Benefits of quitting smoking Review how behavioral activation impacts their overall mood Review avoidance behavior and ruminative thoughts’ significance Strategies for relapse prevention Intersession motivational notification through the App Reinforcement notification of goal achievement through the App	Quitting experience and withdrawal symptoms Discuss and plan for high-risk lapse and relapse situations Motivating factors for maintaining abstinence Benefits of quitting smoking Review how behavioral activation impacts their overall mood Review avoidance behavior and ruminative thoughts’ significance Strategies for relapse prevention
Session 8	Managing the future as ex-smokers Encouragement for abstinence maintenance Support for lapses and relapse Review motivating factors, lifestyle changes, physical and cognitive–behavioral health improvement Review behavioral activation strategies Treatment conclusion and management of potential obstacles Intersession motivational notification through the App Reinforcement notification of goal achievement through the App	Managing the future as ex-smokers Encouragement for abstinence maintenance Support for lapses and relapse Review motivating factors, lifestyle changes, physical and cognitive–behavioral health improvement Review behavioral activation strategies Treatment conclusion and management of potential obstacles

**Table 2 ijerph-19-09770-t002:** App components according to intervention conditions and smoking status during phase 2 (one-year follow-up).

Smoking Status	Experimental Condition (SinHumo App)	Control Condition (Control App)
Abstainer	Smoking self-monitoring tool (to use in case a lapse occurs) Access to the intervention written materials in pdf format Gains and achievements according to abstinence duration after the intervention (health benefits, saved money) Abstinence maintenance tips Self-recorded motivational videos List of reasons to quit to consult in case of experiencing tobacco craving Motivational strategies to maintain abstinence Cognitive and behavioral strategies for coping with cravings Motivational notifications during all the follow-up period	Smoking self-monitoring tool Access to the written materials in pdf format
Smoker (never quit during phase 1)	Smoking cessation tips Smoking self-monitoring tool Possibility of setting a quit date in a calendar Access to the intervention written materials in pdf format Self-recorded motivational videos List of reasons to quit tool Motivational strategies to quit Cognitive and behavioral strategies for coping with cravings Motivational notifications during the follow-up period promoting the initiation of a quit attempt	Smoking self-monitoring tool Access to the written materials in pdf format
Relapser (quit at least 24 h during phase 1)	Smoking cessation tips Smoking self-monitoring tool Possibility of setting a new quit date in a calendar Access to the intervention written materials in pdf format Self-recorded motivational videos List of reasons to quit tool Motivational strategies to initiate a new quit attempt Cognitive and behavioral strategies for coping with cravings Motivational notifications during the follow-up period promoting the initiation of a new quit attempt	Smoking self-monitoring tool Access to the written materials in pdf format

**Table 3 ijerph-19-09770-t003:** Timeline for data collection across the trial.

Measures	Measurement Time-Point				
	Baseline	End of Treatment	3-Months Follow-Up	6-Months Follow-Up	12-Months Follow-Up
Smoking Habit Questionnaire	X				
TUS Tobacco DSM-5 criteria	X				
FTCD	X	X	X	X	X
NDSS	X	X	X	X	X
MNWS	X	X	X	X	X
SSE	X	X	X	X	X
QSU	X	X	X	X	X
BDI-II	X	X	X	X	X
EQ-5D	X	X	X	X	X
APP-Q	X				
CSQ-8		X			
Satisfaction with App questionnaire		X	X	X	X
Follow-up questionnaire			X	X	X

TUS = Tobacco Use Disorder; FTCD = Fagerström Test of Cigarette Dependence; NDSS = Nicotine Dependence Syndrome Scale; MNWS = Minnesota Nicotine Withdrawal Scale; SSE = Self-Efficacy/Smoking Temptation Scale; QSU = Questionnaire of Urgency to Smoke; BDI-II = Beck Depression Inventory II; EQ-5D = EuroQoL-5D Quality of Life Questionnaire; APP-Q = Apps and Smartphone use Questionnaire; CSQ-8 = Client Satisfaction Questionnaire.

## Data Availability

Not applicable.

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
