# Peer review of "A Randomized Clinical Trial to Assess the Efficacy of a Psychological Treatment to Quit Smoking Assisted with an App: Study Protocol"

_ijerph, 2022, doi:10.3390/ijerph19159770_

Round 1
Reviewer 1 Report
This article describes the randomized controlled trial aimed at examining whether the SinHumo smartphone app used as an adjunct to a face-to-face cognitive-behavioural smoking cessation treatment improves cessation rates at the end of treatment, and at 3, 6 and 12 months follow-up, compared to a control group receiving the same smoking cessation treatment and a control App.
The results of this trial may provide a useful tool for improving outcomes in smoking cessation treatments.
Doubts/comments:
1. The recruitment strategy is broad and includes populations that may have different levels of motivation to undertake the treatment. Is the number of patients and their randomization controlled depending on their origin? (Patients referred by their primary care physician who may have another pathology that pushes them to quit smoking are not the same as a person who sees a post on Facebook and thinks it is a good idea to try to quit but does not have the necessary motivation).
2. Is the use of nicotine patches or other methods monitored during the trial?
3. The outcome variables are very much aimed at those who achieve abstinence, but both apps have strategies for 3 groups, Abstiners, smokers and relapsers. What % of reduction in consumption is proposed as an improvement?
4. Figures with Spanish text should be translated so that readers could better understand the structure and contents of the SinHumo and Control Apps interfaces.
5. To avoid confusion, in figures (and text) where ‘App’ appears, it should always be specified whether it refers to SinHumo App or Control App.
6. In tables 1 and 2 you could also specify Experimental condition (NoSmoke App) Control condition (Control App).
Congratulations, I look forward to reading the results soon.
Reviewer 2 Report
The authors have done an amazing job by developing a user friendly app to address an important and valid public health concern. However, I would suggest few additions:
1. If you could add details on content development process for this app.
2. If you could elaborate CBT model used for this app.
3. If more detail on data analysis could be added.
4. If you could add outcomes of the app in detail in relation to the tools used.
